



# The Role of Point Discharge in the Development of Atmospheric Electricity

Blair P. S. McGinness[1], R. Giles Harrison[1], Karen L. Aplin[2], and Martin W. Airey[1]

[1]Department of Meteorology, Earley Gate, University of Reading, Reading, RG6 6ET, UK
[2]Queens Building, University Walk, Bristol BS8 1TR, UK

**Correspondence:** Martin W. Airey (m.w.airey@reading.ac.uk)

**Abstract.** Point discharge, like lightning, is an atmospheric electricity process which has been observed directly and indirectly for centuries. Point discharge occurs when an electric field is enhanced at a point, causing local ionisation of the air and allowing a current to flow between the object and atmosphere. Point discharge sensors are simple instruments which measure the discharge currents caused by enhancements of the atmospheric electric field. In the early 20th Century, several milestone

atmospheric electricity investigations were performed which employed the effects of naturally occurring point discharge currents and the measurements made by point discharge sensors. Point discharge was central to some of the arguments made in the proposal of the global atmospheric electric circuit, and the early evidence found to support this model. Point discharge sensors continued to be used throughout the 20th and 21st centuries, with understanding of their operations being developed further in this time.

# 1 Introduction

Through dramatic processes, such as lightning strikes, it can be said that atmospheric electricity has been observed for all of human existence (Schlegel, 2024). The understanding of these processes developed more recently, however, with the conceptual picture of the global atmospheric electric circuit being proposed in the early 20th Century. Central to many of the investigations driving this understanding was the process of point discharge. This occurs naturally, acting as an important current flow

in the global atmospheric electric circuit, but dedicated point discharge sensors can also be used as atmospheric electricity instruments.

     Point discharge occurs when an electric field is enhanced at the tip of a sharp point. The local enhancement of the electric field at these points leads to the air becoming ionised via charged particle collisions. This region of ionisation then allows a current to flow between the object and the surrounding air, referred to as a corona discharge or point discharge current (PDC).

These currents can sometimes lead to a glow visible to the naked eye.

     Through this visible glow, the process of point discharge has long been observed. In the 1st Century BCE, De Bello Africo, attributed to Julius Caesar, described an occurrence of the phenomenon as the tips of spears "spontaneously caught fire" (Caesar, c.40 BCE [1955]). In the 1st Century CE, Pliny the Elder described his understanding of the phenomenon, stating that there was a "luminous appearance" which occasionally attached to javelins and parts of ships (Pliny the Elder, 77 [1855]). Similarly





to many early lightning observations, these reports predate understanding of the electrical nature of the point discharge process, instead attributing it to the work of the gods. Many more observations of the phenomenon have occurred since, with it often being observed on the masts of ships (Von Engel, 1965; Chalmers, 1967; Wescott et al., 1996). The associations with ships, along with it's flame-like appearance, have led to the visual phenomenon being named "St. Elmo's fire" after the patron saint of sailors, St. Elmo (Perkins, 1979; Wescott et al., 1996). St. Elmo's fire has also been observed to occur from sharp tips on aircraft (Wescott et al., 1996; Mulvey et al., 2017).

In the 18th century, several investigations identified the presence of electrical charge in the atmosphere. These discoveries are best known through Benjamin Franklin's "kite experiment", which was able to draw sparks from a kite flying in a thunderstorm (Franklin, 1752). At similar times to this experiment, however, other investigations showed that electrical charge was present in the atmosphere in fair weather conditions (Le Monnier, 1952; Canton, 1753; Mazeas and Parsons, 1753). By identifying the presence of electricity in the atmosphere, these investigations were a fundamental step towards our current understanding of atmospheric electricity. Franklin is also known for his work investigating lightning rods. As noted in his minutes dated November 7 1749, Franklin suggested that St Elmo's fire would be able to be produced from "electrical fire" being drawn from a cloud into a lightning rod (Van Doren, 1938). The identification that St. Elmo's fire is electrical in nature was a significant development in our understanding of the phenomenon.

The process of point discharge can be used to investigate the local atmospheric electric field. The polarity and magnitude of a discharge current are dependent on the properties of the electric field inducing the current. Point discharge sensors are instruments designed such that they have a sharp point which enhances the atmospheric electric field and causes point discharge to occur. This discharge current is then recorded by a meter. An example of a point discharge sensor deployed at the Reading University Atmospheric Observatory is shown in Fig. 1. To relate the point discharge measurement from such a sensor into an observation of the atmospheric electric field, it is important to have a description of the operation and geometry of the sensor. A history of various parametrisations that were developed for PDC sensors is given in Sect. 4. Point discharge sensors are typically very simple devices, giving them numerous advantages over more complicated atmospheric electricity instruments, such as field mills. These advantages will be considered further in Sect. 3.4.

Point discharge has been important to many atmospheric electricity investigations performed in the 20th and 21st centuries. PDC sensors have been used in a range of capacities, from taking observations of the potential gradient (PG) to making estimates of the vertical current flow at a given location. Arguments involving the effects of naturally occurring point discharge currents have also been made, without direct use of point discharge sensors. In addition to their usage in historical investigations, point discharge sensors continue to be used as atmospheric electricity instruments. Modern understanding of these sensors may allow some historical data to be reanalysed. Consequently, there is a clear motivation to have a documented account of the history of investigations involving point discharge.

A history of several of the investigations featuring point discharge are described in the subsequent sections, first considering several investigations important to the development of key atmospheric electricity concepts from the 20th Century, then considering a number of subsequent investigations, and finally several investigations which furthered the understanding of the operation of PDC sensors.

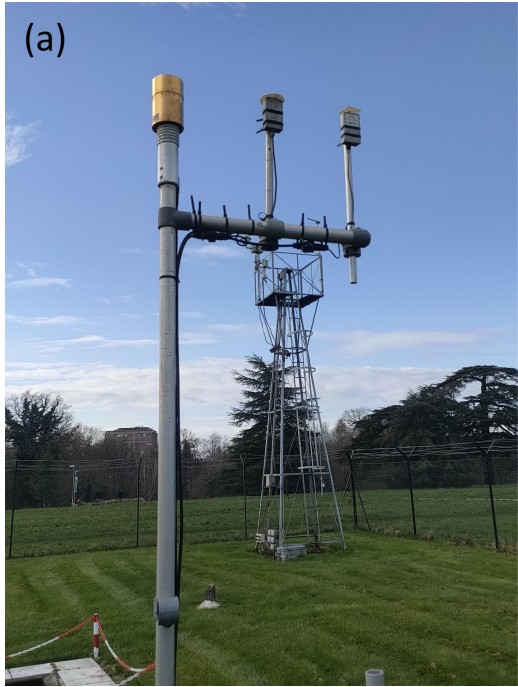
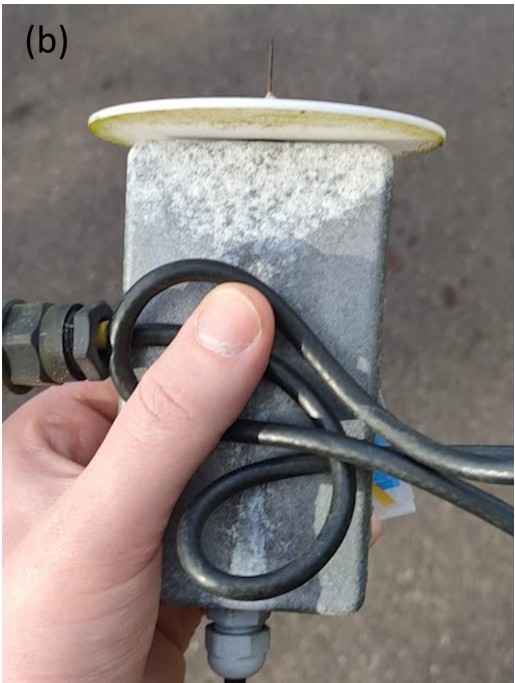

**Figure 1.** Photographs of point discharge sensors deployed at the Reading University Atmospheric Observatory. (a) shows the top of a 3m tall mast, with three atmospheric electricity sensing devices mounted on it. The leftmost of these devices is a Chubb JCI 131 field mill, with the two rightmost devices being point discharge sensors. (b) shows a close up view of one of these point discharge sensors. This photo was taken following the sensor being deployed in the field site for at least 3 years, and some weathering is visible on the device; in particular, the sharp point at the tip of the sensor is showing visible rusting. Despite this weathering, the point discharge device was continuing to work well in this deployment.

## 2 Atmospheric electricity in the early 20th century

By the early 20th Century, there had been a great advancement in the knowledge of atmospheric electricity. It was known that the surface of the Earth was negatively charged with respect to the atmosphere, with further observations that the PG at the surface was typically positive (Exner, 1900). Further, it was known that a charged conductor in the air would lose this charge through dissipation in a manner consistent with the air being conductive, and it had been observed that ions were present in the atmosphere (Exner, 1901; Wilson, 1901). These results together introduced a problem, however. Given that the atmosphere was conductive, the charge held by the Earth should dissipate readily, as for other charged conductors in air. A further uncertainty was present regarding the charge structure of thunderclouds, despite their electrical nature having been known for a substantial time.

To explain why the charge of the Earth was persistent, several theories were considered: It was suggested that as negative ions in air were more mobile than positive ions, the Earth would undergo collisions with them more readily and would become



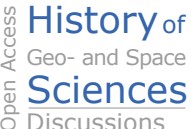

negatively charged, while the air becomes positive (Exner, 1901; Wilson, 1903). Alternatively, it was suggested that electrons emitted by the Sun could pass through the atmosphere, colliding with the surface of the Earth, causing it to acquire a negative charge (Wilson, 1903). Another suggestion, which started the journey towards the conception of the global atmospheric electric circuit model was that in fair weather conditions the positive PG would lead to a conduction current which acted to dissipate

the negative charge of the Earth, however, in areas of precipitation a negative charge would be brought down by charged rains, restoring the Earth's negative charge (Wilson, 1903).

In was stated by Wilson (1906) that in order to judge various atmospheric electricity theories it would be necessary to have measurements of the conduction current between the atmosphere and Earth at various locations. Wilson further noted that it would be greatly desirable to have simultaneous observations of the charge brought down by precipitation, no doubt in an

attempt to verify his theory that precipitation caused the Earth to charge negatively. To facilitate these measurements, several experiments testing a method of observing the conduction current were reported by Wilson (1906, 1908).

Observations on the charge carried by thundercloud rain were later reported by Simpson (1909). These measurements were taken by recording the potential transferred to a galvanised iron "receiver" collecting rainfall. The rain would then run off of the receiver into a tipping bucket rain gauge, measuring the rainfall rate. Together, these two measurements allowed the

charge per cubic centimetre of rain to be determined. Simpson found that there was substantially more positive charge brought down by rain than negative charge, in contrast to the predominant negative charge which would be required for the theory that precipitation maintained the negative charge of the Earth. Later, Simpson (1912) discussed subsequent observations occurring from a range of other locations, showing conclusively that for all types of rain a positive charge, not negative, is brought to the Earth.

It was clear that from the results from Simpson (1909) that further theory was required to explain the persistence of the negative charge of the Earth's surface. Point discharge played a key role in a number of the investigations which attempted to gain this understanding.

## 2.1 Controversy over thundercloud polarity

Related to the precipitation hypothesis for sustaining the Earth's negative charge, there was the apparent assumption that the

charge structure of thunder clouds was that of a positive dipole (i.e. a positive charge situated above a negative charge). The result of Simpson (1909) implied the opposite, however; the fact that the rain underneath thunderclouds was predominantly positively charged implied that the base of these clouds was positive, and so suggested a negative dipole (i.e. a negative charge above a positive charge). As there had been no direct observations of the charge structure, however, there continued to be debate on the polarity of thundercloud dipoles throughout the early 20th century. A review of this disagreement between two

scientists, G.C Simpson and C.T.R. Wilson has been given by Williams (2009).

Several years after his previous precipitation theory had been disproven, Wilson (1921) reported his findings of an investigation into the electric field during lightning strikes. This investigation found that the electric field changes observed during the majority of lightning strikes were consistent with negative charge being brought from the cloud to the Earth, which Wilson

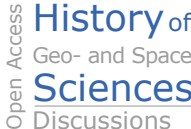

named a "positive discharge". Wilson used arguments based on the relative frequency of these positive discharges, to suggest
the the presence of a positive dipole in thunderclouds.

In this report, Wilson addressed the evidence suggesting a negative dipole found from the previous work by Simpson. Wilson
argued that the arrival of positively charged rain at the Earth's surface did not provide evidence that the rain was positively
charged as it left the cloud, instead suggesting that positive ions emitted from the surface could cause a reversal of the polarity
of the rain droplets. This reversal of polarity would cause the droplets to be positively charged when observed at the surface,
despite being negatively charged when at the cloud base. Wilson suggested point discharge currents emitted from blades of
grass and the leaves of trees could be sources for these positive ions. The general upwards flow of positive ions from point
discharge processes, along with the negative charges carried downwards from the cloud by the rain would together lead to a
negative charge being brought to Earth by the thundercloud. Coupled with a current flow above the clouds, bringing a positive
charge to the upper atmosphere, this would act to maintain the charge separation between the surface and ionosphere; restoring
the potential difference which is lost via the fair weather conduction current. These mechanisms formed the basis of Wilson's
model of the global atmospheric electric circuit, a concept which continues to offer explanatory value in atmospheric electricity.

## 2.2 Evidence for the global circuit model

Following the proposal of a global atmospheric electric circuit by Wilson, Whipple and Scrase (1936) reported their results
of an investigation into the point discharge data recorded at Kew Observatory. The PDC data used in this investigation was
measured using a galvanometer connected to an sharp point on the tip of a tall mast. Whipple and Scrase analysed both the
PDC response of the sensor to atmospheric conditions, developing a parameterisation describing the operation of the sensor,
and several trends recorded by the PDC sensor in a period of continuous observation. This parameterisation was important to
the development of PDC sensors as atmospheric electricity instruments, as it allowed the measured currents to be converted
into a measure of the PG. This parameterisation, and the development of subsequent parameterisations will be discussed in
Sect. 4.

In their observations of the continuous operation of the PDC sensor, Whipple and Scrase observed that there was a diurnal
variation in the net outflow of positive discharge currents from the sensor. Based on previous suggestions that thunderstorm
variations were responsible, the PDC variations were compared against the diurnal variation of global thunderstorm frequency.
The figure used by Whipple and Scrase for this comparison is shown in Fig. 2. From this comparison, it was seen that there
was a close relationship between the variations of global thunderstorm area and local PDC.

Whipple and Scrase further went on to compare the diurnal variation in thunderstorm area with the PG data recorded by the
ship *Carnegie*. This dataset, often referred to as the "Carnegie curve", shows a diurnal variation in PG dependent on Universal
Time, not local time (Harrison, 2013, 2020). Whipple and Scrase found that there was a good agreement between the Carnegie
curve and the global thunderstorm area, suggesting that the two were related. This, along with the variations in PDC, helped
to confirm that disturbed weather regions on Earth were in fact linked to fair weather regions, and has been considered an
important milestone in the history of atmospheric electricity, offering confirmation to Wilson's theory of a global atmospheric
electric circuit.

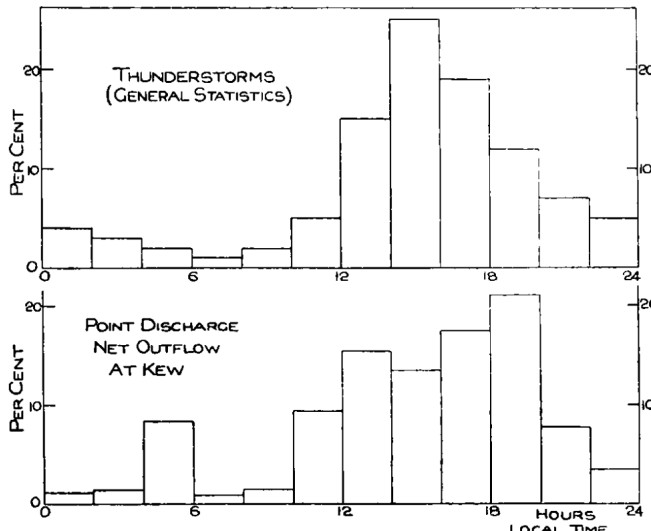

**Figure 2.** Comparison of the diurnal variation in thunderstorm occurrence with the net outflow of point discharge at Kew Observatory. Figure taken from Whipple and Scrase (1936).

### 2.3 Observation of thundercloud polarity

The confirmation of the global circuit model did not definitively provide evidence for a given model of the charge structure of thunderclouds, so did not end the dispute between Wilson and Simpson. To resolve this dispute, in situ observations of the charge structure were desired. To make these observations, Simpson and Scrase (1937) developed a balloon borne instrument for measuring the PG, which they named the alti-electrograph[1]. The alti-electrograph observed the polarity of the PG as it rose through the atmosphere, while additionally taking measurements of the atmospheric pressure and humidity. To measure the PG polarity, the alti-electrograph made use of point discharge electrodes extending above and below the instrument. It was known that for a long conductor in an electrical field, point discharge will occur such that a conventional current will flow into the conductor at the end where the atmospheric potential is positive, and flow out from the conductor at the end where the potential is negative. The alti-electrograph was constructed such that the electrodes formed a tall vertical conductor, and measurements were taken of the polarity of current flowing through the two electrodes, which were used to infer the polarity of the PG. The polarity of current was determined by connecting both electrodes to a piece of pole finding paper. As the current flowed between the electrodes, a deposit of Prussian blue was built up at the anode, with no such marking at the cathode. The current polarity was determined as a function of time by moving the electrodes across the paper. Finally, by recovering the alti-electrograph after its sounding, it was possible to compare the polarity data against the pressure data, providing a profile of the polarity with height. A system was designed to release the alti-electrograph instrumentation from the balloon at a given

---

[1]The mixed Latin and Greek roots of this word were excused as Simpson and Scrase wished to convey the meaning of "height" readily in the instrument's name

History of Geo- and Space Sciences Discussions Open Access

altitude, and have it descend to Earth using a parachute. Typically, this separation was set to occur at an altitude of 8-9 km, however in some cases this separation did not occur and measurements were taken even higher than expected. A photograph of the alti-electrograph as shown in the report by Simpson and Scrase has been included in Fig. 3.

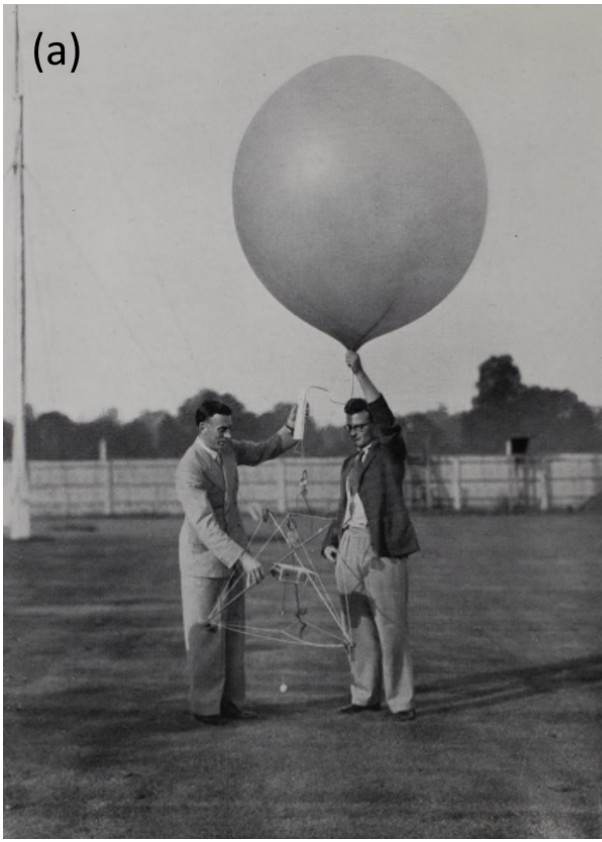
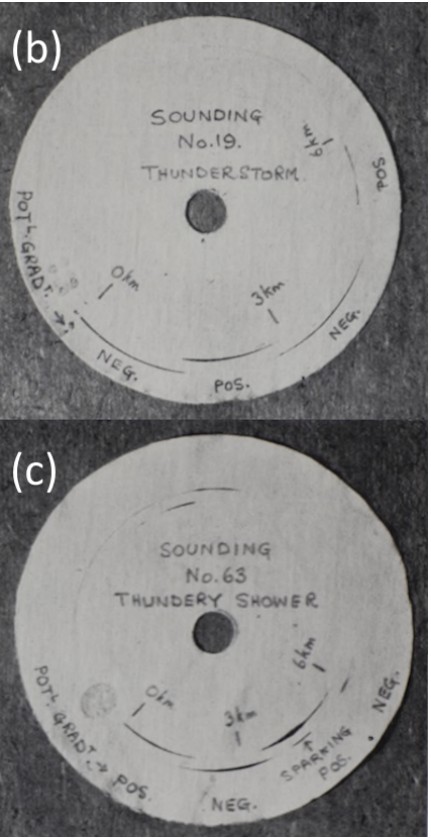

**Figure 3.** Photographs of the alti-electrograph, and the data recorded by it. (a) shows the alti-electrograph being launched by Sir George Clarke Simpson (left) and presumably Frederick John Scrase (right). (b) and (c) show the data recorded by the alti-electrograph instrumentation from two different deployments. The altitudes of 0, 3, and 6 km have been annotated on these records, along with the polarity of the recorded PG as "pos" or "neg". Additionally, in (c), a burn mark from electrical sparking has been indicated. The "striking" blue colour of the original record is lost in this black and white photograph; in the originals the data recorded in blue was clearly distinct from the burn mark. Figures have been edited from Simpson and Scrase (1937).

The alti-electrograph apparatus was used in a number of atmospheric soundings; from July 1934 to October 1936 70 of these soundings were performed, in a range of weather conditions. Through analysis of trends in the data, it was found that the main body of thunderclouds were negatively charged, with a positively charged upper region. The observation of positive dipoles in these thunderclouds brought about an end to the disagreement between Wilson and Simpson on the charge structure of thunderclouds. Additionally, the positive charge at the upper edge of the clouds supplied a mechanism for the charge





separation between the surface and ionosphere to be maintained. This provided additional evidence for the global circuit model theorised by Wilson.

From the alti-electrograph data, it was found that frequently an additional smaller positively charged region was located at
165 the base of the cloud, below the negatively charged region. This finding is in agreement with our modern understanding of thunderstorm charge distributions; it is known now that in a typical thundercloud, there are two regions of net positive charge, a smaller one at the base and larger one in the upper cloud, with a region of negative charge in between (Williams, 1989). This arrangement is referred to as an electrical tripole. The agreement with our modern understanding shows the high quality of the measurements made with the alti-electrograph device.

As discussed previously, Simpson (1909) argued in favour of a positive dipole in thunderclouds, due to positively charged rains observed at the surface. It is known now that the source of these charged rains is the small positive region at the base of thunderclouds. Before this positive region was discovered, Wilson made arguments for a negative cloud base leading to positively charged rains, owing to positive ions emitted from PDCs at the surface. It is important to note that although it is now known that these emitted ions are not required to explain the polarity of charged rain, it is still considered that the effects
of PDCs on the surface are important to the flow of charge in the global atmospheric electric circuit. A number of studies published following the discovery of the tripolar structure of thunder clouds have drawn importance to the effects of these PDCs, with them identified as an important method by which a net negative charge is brought to the Earth's surface from the atmosphere (Immelman, 1938; Chalmers, 1967; MacGorman and Rust, 1998).

## 3 Other PDC investigations of the 20th and 21st Centuries

The investigations described so far have demonstrated the critical role that point discharge has played during an important period of atmospheric electricity history. Following the confirmation of the global circuit model, point discharge sensors continued to be used for a range of atmospheric electricity investigations, a number of which are highlighted here.

### 3.1 Alti-electrographs and Coronasondes

The initial alti-electrograph investigation by Simpson and Scrase, showing the positive dipole in thunderclouds, was well
accepted. Despite this, the finding of lower positive regions of charge in these clouds met with difficulty. In a subsequent paper, Simpson and Robinson (1941) discuss the complaints against this finding by re-analysing the data from the original soundings, again finding evidence for this lower charge region. In addition to this re-analysis, multiple further soundings using the alti-electrograph were performed by Simpson and Robinson. In all of these soundings, the lower positive charge was identified, providing strong evidence for its existence. In these soundings, Simpson and Robinson additionally attempted to determine the
nature of the charge separation process creating the positive dipole, by comparing the estimated temperature with the charge structure. Through this analysis, it was determined that the centres of the main charge regions were both below 0°C, suggesting that processes involving ice crystals were important to this charge separation.





To investigate the electrical structure of thunderclouds further, other point discharge instrument packages were developed, improving upon the alti-electrograph instrument. As mentioned previously, the alti-electrograph required the recovery of the instrumentation to retrieve the collected data. A subsequent investigation by Belin (1948) managed to remove this limitation by modifying a radiosonde to take point discharge measurements. The radiosonde would then transmit the data to a receiver on the surface, removing the need to recover the instrumentation.

Later, Chapman (1956) used a similar (radiosonde-based) method to investigate the structure of thunderclouds. The radiosonde instrumentation was used to investigate how the electrical structure aligns to the temperature profile inside these clouds, since virtually all previous investigations had not measured the temperature and the PG directly. These modified radiosondes were additionally used to investigate the PGs present inside blizzards (Chapman, 1953). Through this investigation, Chapman was able to show that the electrification in these snowstorms was not limited to the ground.

Recognising the value of these previous studies, Weber and Few (1978) developed a similar instrument. The authors described such an instrument, using a radiosonde modified to take point-discharge measurements, as a "coronasonde". The coronasonde was described as inexpensive, easy to use, and was able to provide quantitative information on the PG inside electrical clouds. Later, in order to ensure the accuracy of this quantitative information, laboratory experiments were performed to understand the influence of several atmospheric parameters - such as temperature, pressure, water vapour content, and wind speed - on the recorded point discharge currents (Byrne et al., 1986). The utility of the coronasonde instrument meant that it was used in a number of investigations (Weber and Few, 1978; Weber et al., 1983; Byrne et al., 1983). These investigations helped to further the understanding of the charge structure inside thunderclouds, as well as provide more evidence for the presence of an electrical tripole.

### 3.2 Non-balloon soundings

Atmospheric soundings using point discharge are were not limited to balloon based measurements; A NOAA technical report, and later patent, describe a rocket-borne instrumentation package for taking point discharge measurements (Ruhnke, 1971, 1973). A diagram of this instrumentation is shown in Fig. 4. It was observed that ground based observations of the PG in clouds were difficult if not impossible to obtain. Additionally, although aeroplane and balloon based observations allowed the PG to be measured in situ, there were a number of downsides to these observation methods, such as the requirement for aeroplanes to penetrate the cloud several times, and the slow ascent speed of balloons. As an alternative, rocket-based observations were suggested, as a way to rapidly obtain a profile of the PG through the cloud. It was noted that the high speed of the rocket meant that rapid measurements would have to be obtained, in order to have a good spatial resolution of datapoints; a sampling rate of 25 Hz was suggested as a requirement. Additionally, the sensor would have to be able to survive large accelerations (up to 50 g). These criteria led to the selection of a point discharge sensor to measure the PG.

PDC sensors have also been used onboard spacecraft. The Venera 13 & 14 missions utilised point discharge sensors on board their "groza-2" instrumentation package, taking electrical measurements in the atmosphere of Venus (Ksanfomality et al., 1982). Little information is available on the specifics of these sensors, however the principal investigator of the instrumentation package, Leonid Ksanformality, confirmed during a personal communication with Lorenz (2018) that it contained a point


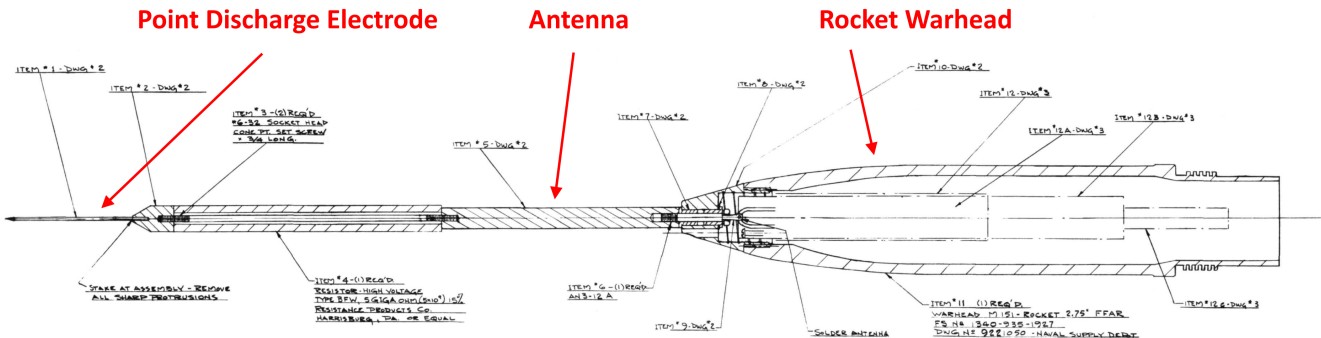

**Figure 4.** Annotated drawing of the rocket borne point discharge instrumentation described by Ruhnke (1971). This drawing was part of a series of diagrams showing the construction of the instrumentation package. Several additional annotations have been indicated in red, showing the location of the point discharge electrode, antenna, and the rocket warhead. The point discharge electrode was constructed from a tungsten steel rod of diameter 1/16th inch, sharpened to a point. The instrumentation package was designed for use with the FFAR rocket (nicknamed "mighty mouse"). Figure has been altered from Ruhnke (1971).

discharge electrode. Some recent work has been performed recently to attempt to infer details on the sensor design via logical and engineering considerations (Sukanjanajtee and Aplin, 2024).

### 3.3 Grounded investigations; the balance of charge

One of the most prolific point discharge researchers in the 20th Century was J.A. Chalmers, who authored 17 papers on the subject between 1941 and 1967, with the final paper being published posthumously (Aplin, 2018; Jhawar and Chalmers, 1967). Chalmers was instrumental in the development of parametrisations of the operation of point discharge sensors, which will be discussed further in Sect. 4. In addition to these investigations of PDC sensors, much of Chalmers' research considered the balance of charge in the global atmospheric electric circuit, with a focus on the contributions that naturally occurring point

discharge currents had on the flow of charge in disturbed weather regions. To understand the quantity of naturally occurring point discharge, many investigations were performed by Chalmers and others, considering the point discharge occurring from trees.

An investigation by Schonland and Wilson (1928) considered the relative importance of various current sources between the atmosphere and Earth in disturbed weather regions. The current flows of lightning strikes, charged rain, and PDCs were

considered in this investigation. To estimate the amount of PDC caused by a thunder cloud, the current flowing through a single tree was investigated. For this experiment, a thorn tree (Acacia Karroo) was cut down and mounted on an insulated platform. The current flowing through the tree was then measured for various electric field strengths observed under a thundercloud. As the tree had been cut down for this experiment, it was noted that the leaves withered. The thorns on the tree remained, however, and additional branches were occasionally attached. From the results of this experiment, an estimate was produced for the



amount of PDC caused by a thundercloud. This estimate was compared against similar estimates made for the current flow caused by lightning and charged rain, where it was concluded that point discharge was the dominant process for the transfer of charge to the surface.

As pointed out by Maund and Chalmers (1960), the experiment performed by Schonland and Wilson was not representative of the currents flowing into a living tree. Citing another unsuccessful investigation of the discharge current flowing through a

tree as motivation, Maund and Chalmers set out to investigate the discharge current through trees through indirect methods. In conditions with appreciable wind speed, the ions emitted from a point discharge source are expected to travel with the wind. It is thus expected that a reduction in the PG will occur downwind, due to the space charge produced by the presence of these ions. Maund and Chalmers developed theory to determine the expected reduction in PG for a given discharge current. This theory was verified through taking measurements upwind and downwind of a discharging point located atop a tall mast. Maund

and Chalmers then applied this to indirectly measure the point discharge from a tree, through measurements of the reduction in downwind PG. The results from this investigation appeared to imply that a tree in leaf produced significantly less point discharge than previous estimates suggested.

A subsequent investigation by Bent et al. (1965) attempted to experimentally measure the space charge produced by point discharge from trees. From this investigation it was shown clearly that a space charge was produced, with a similar magnitude

to that from an artificial point. This investigation concluded that previous estimates of the total point discharge rate were likely accurate, and that point discharge plays an important role in the flow of negative charge to the surface of the Earth in disturbed weather regions.

A number of investigations have attempted to directly measure the current flowing through a living tree, using electrodes placed inside the tree (Kirkman, 1956; Milner and Chalmers, 1961; Chalmers, 1962b, 1964; Jhawar and Chalmers, 1967).

The general finding from these investigations was that the results from such a method were inconsistent. Chalmers (1964) concluded that the tree investigated did not behave as a simple electronic system, and could not be represented by a resistor-capacitor system.

Building upon previous investigations, Jhawar and Chalmers (1967) took measurements of the current flowing through a live tree planted in a pot. The tree was placed between two metal plates, which were charged such that a voltage up to 50 kV

was able to be produced between them. The point discharge flowing into the tree, towards Earth, was then measured using a galvanometer. This allowed a relationship between the voltage and point discharge current to be determined for the case of a single, entire tree. The laboratory set up, including the spruce tree, is shown in Fig. 5. From measurements of the current at different voltages, a relationship was able to be identified between the applied voltage and point discharge current. This relationship was of the form:

$$I = V(V_0 - V)^2 \tag{1}$$

where I is the point discharge current through the entire tree, V is the voltage, and $V_0$ is the minimum voltage for point discharge to occur. This equation was further derived theoretically in the same paper.

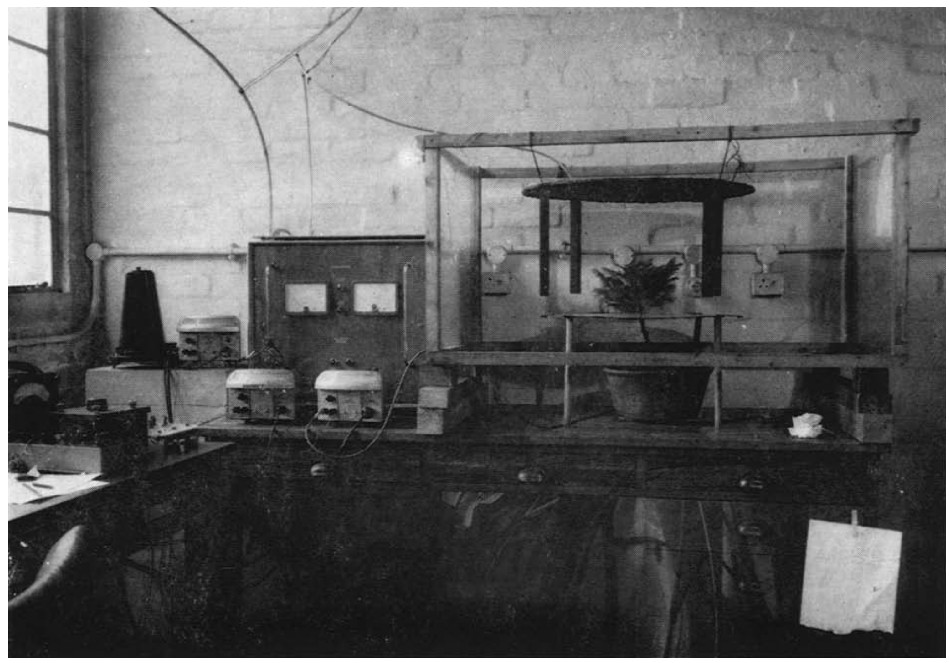

**Figure 5.** Experimental set up from Jhawar and Chalmers (1967). A live spruce tree, approximately 60 cm tall, in a plant pot is positioned between two metal plates. The metal plates are charged to produce a voltage between them. The large box to the left of the tree appears to be the power supply for this, with a cable visible running towards the upper plate. The total point discharge flowing into the tree was measured using a galvanometer connected between the plant pot and the earth. The three smaller boxes located next to the power supply appear to be optical galvanometers, with the rightmost of these wired up to the plant pot. Measurements of the point discharge were taken for a range of different voltages, on several days. Figure taken from Jhawar and Chalmers (1967). Reproduction rights were acquired for this figure.

These investigations have verified that naturally occurring point discharge is of some import. The nature of the point discharge process occurring from trees has been investigated in a range of capacities, and it has been shown that point discharge has an important role in the balance of current in the global atmospheric electric circuit.

### 3.4 Modern applications of PDC sensors

Point discharge sensors may be considered historical atmospheric electricity instruments, however they still have some relevance today. As mentioned in Sect. 1, PDC sensors are very simple devices which lack any moving parts. As such, these sensors tend to be very cheap and robust instruments, which require little maintenance. This has been demonstrated to be advantageous to the deployments on air and spacecraft discussed so far. In addition to this, PDC sensors are well suited to grounded deployments in remote or harsh conditions where damage to delicate components would be likely, or regular maintenance is not possible. An example of this utility is demonstrated by an investigation of the electrical properties of dust devils by Lorenz et al. (2016). In this investigation, an instrument package was deployed in the Chihuahuan desert, and left to collect data for over a month without maintenance. Given the harsh conditions a point discharge sensor was used to take electrical measurements. It





was found that the point discharge sensors operated well in their deployment, providing electrical measurements of the dust devils which passed over the instrumentation.

Across fair weather and disturbed weather conditions, the PG can vary across several orders of magnitude. To take measurements across this entire range, it is necessary to have instrumentation with a very large operational range. As is evident from the parametrisations that will be discussed in section 4, a range of several orders of magnitude in the PG will also cause a range of several orders of magnitude in the current recorded by a PDC sensor. One method developed to capture the full range of these currents is to have PDC instrument with a logarithmic response. Such a sensor has been described by Marlton et al. (2013). This device uses LEDs as feedback resistors to produce a logarithmic electrometer, providing an output voltage proportional to the logarithm of the input current. The wide range of this instrument allows atmospheric electricity observations to be made across both fair weather and disturbed weather conditions, without need of any range-switching.

Point discharge can be an electrical hazard, making it important to monitor. Burt (2022) describes an event where anomalous measurements were recorded by the data logger at the Reading University Atmospheric Observatory. It was verified, through comparison with data recorded by the point discharge sensor, that these anomalous readings were caused by point discharge occurring from the anemometers. Without having a point discharge sensor at the field site, which measured large discharge currents at the same time as the anomalous readings, it would have been difficult to identify the source of the anomaly. As such, the presence of a point discharge sensor is demonstrably advantageous to such a field site.

## 4    Characterising the response of PDC sensors

It has been understood for a while that the magnitude and polarity of a point discharge current is dependent on the electric field driving the current. What has not been understood, however was the exact form of the relationship between the discharge current and the electric field (and any other atmospheric parameters). An accurate description of the operation of these sensors was desired, since it would allow PDC sensors to be used effectively as atmospheric electricity instruments. As such, there have been a number of investigations performed which attempted to obtain such a parameterisation, through both theoretical and empirical means; many of the theoretical investigations were performed by J.A. Chalmers, who was noted as being exceptionally driven in finding theoretical analytical solutions to real-world problems (Aplin, 2018). A number of the key investigations attempting to describe the operation of PDC sensors are now described.

As was mentioned in Sect. 2, an early attempt to describe the operation of a PDC sensor was performed by Whipple and Scrase (1936). This investigation involved analysis of the PDC data recorded at Kew Observatory, comparing it against PG data. From this empirical investigation, a power law relationship between the PG and PDC was identified, taking the form:

$$I = a(F^2 - M^2) \tag{2}$$

where $I$ is the discharge current, $F$ is the potential gradient, $M$ is a constant describing the minimum PG for point discharge to occur, and $a$ is some other constant. This equation was later derived by Chalmers (1952) from theoretical considerations.





For this derivation, it was assumed that the point discharge process was limited by the build up of space charges, and that these charges were removed from the vicinity of the sharp point by electrical forces alone.

Chalmers' 1952 derivation made the assumption that wind had a negligible effect on the point discharge process. Note that the literature makes the point that this assumption is not the same as the assumption that the wind speed itself is negligible;
the effects of the wind speed on the space charge could also be assumed to be negligible if we are dealing with several PDC sources close together (Chalmers, 1952; Kirkman and Chalmers, 1957). In this case, as the ions forming the space charge are removed from the vicinity of a given point, the ions from another point's space charge move to replace them. As such, the wind speed does not act to greatly reduce the space charge around a given point.

Subsequent investigations have found that the assumption that the effects of wind speed are negligible is not valid in many
circumstances. The form of the equations derived including the effects of this wind speed are variable, however. Through theoretical considerations, Chalmers and Mapleson (1955) derived an equation to describe the PDC from an isolated point. This characterisation was given by:

$$I = a(hF)^b W^c \tag{3}$$

where $I$, $F$ are as before, $h$ is the height of the point above ground, $W$ is the wind speed, and $a$, $b$, $c$ are constants (with
theory providing $b + c = 2$). Note that it was identified at the time that since there is no term dependent on the minimum PG for PDC to occur, as there is in Eq. 2, then this equation will only be approximately accurate, and will diverge from observations at low PGs. To verify the theoretical equation, Chalmers and Mapleton investigated the PDC detected by a captive balloon. The data found from this investigation reinforced that the form of equation derived was valid.

A further series of investigations by Chapman (1956), Large and Pierce (1957), Kirkman and Chalmers (1957), and sugges-
tions by Chalmers led to another form of characterisation including wind speed being developed. This was given in terms of the electric potential at the discharge tip rather than the potential gradient. Converting to potential gradient, assuming a constant PG between the surface and the discharge point, allows the description of the PDC sensor to be written as:

$$I = a(F - M)(W^2 + bF^2)^{1/2} \tag{4}$$

where $I$, $F$, $W$, $M$, are as before, and $a$, $b$, are constants. This form of equation was further investigated by Chalmers (1957);
Chalmers used data from a number of previous investigations to investigate if this equation was able to describe the operation of a PDC sensor in general. It was found that if the parameters $a$, $b$, and $M$ were adjusted, then for each dataset investigated, a good fit was able to be obtained. Later, via theoretical considerations, Chalmers (1962a) was able to derive this equation from theoretical considerations. This derivation provided some theoretical meaning to the constants $a$, $b$. The values found from these theoretical considerations were not in exact agreement with those found empirically, however.

In addition to the parameters discussed so far, lab based investigations of point discharge have shown further dependencies on properties such as the temperature, gas pressure, and atmospheric composition (Haidara et al., 1997; Bologa et al., 2011).

Typically at the surface of Earth these values remain fairly constant, so the parameterisations which neglect them will remain mostly valid. It does raise the point, however, that these parameterisations are not describing the full process of the operation of the PDC sensors, and the inclusion of additional terms would likely increase their accuracy. Recently, in an investigation of the 355 operation of a point discharge sensor with relatively rapid sampling (1Hz), it was observed by McGinness et al. (2024) that the sensor was sensitive to both free currents (caused by the flow of charge), and displacement currents (caused by a time-varying electric field). As such, it was found that including additional terms dependent on the rate of change of the PG leads to an improvement in the accuracy of the description of the PDC sensor. To describe the operation of PDC sensors accurately, it may be necessary to identify and include further terms in such a description. If this were possible, and the operation of the 360 sensors were able to be described with a high precision, then it would greatly improve the utility of these sensors as accurate atmospheric electricity instruments; it has previously been observed that a lack of a good description of the operation of PDC sensors has limited their utility as atmospheric electricity instruments (MacGorman and Rust, 1998).

## 5 Conclusions

Point discharge has played an important role in atmospheric electricity history, particularly throughout the 20th and 21st 365 centuries. Both through considerations of the naturally occurring process, and from measurements from point discharge sensors, point discharge has played an important role in several milestone investigations of 20th century atmospheric electricity. These investigations include considerations made by C.T.R Wilson in proposing the global circuit model, some early proof of this model as identified by Whipple and Scrase, and the first in situ measurements of the electrical structure of thunderclouds.

Point discharge sensors have proved to continue to be effective atmospheric electricity instruments in the years following 370 this, being used in many other investigations. These investigations involved grounded and airborne sensors, both in Earth's atmosphere and the atmosphere of Venus. During this time, knowledge of the operation of PDC sensors has been advanced by the development of different characterisations of their operation. The advancement of these characterisations may allow PDC sensors to be even more accurate as atmospheric electricity instruments in the future.

*Author contributions.* BM, GH, KA, MA contributed to the ideas. BM produced the original figures, and performed the edits to pre-existing 375 figures. BM prepared the original draft, with edits suggested by GH, KA, MA.

*Competing interests.* Some authors are members of the editorial board of History of Geo- and Space Sciences.

*Acknowledgements.* Much of this research was performed during the undertaking of a PhD project, funded by STFC.

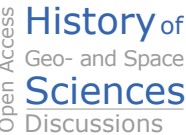

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
