# Peer review of "The Role of Point Discharge in the Historical Development of Atmospheric Electricity"

_History of Geo- and Space Sciences, 2025_

## Author Response (AR1)

**Reviewer #1 Comments**

| I wonder whether it would be good to use the data shown in Fig. 2 to plot a graph, having twelve points, showing PDC (y-axis) against Thunderstorms (x-axis), The correlation coefficient could be calculated, and the significance of the result assessed. | Additional analysis has been performed, with a scatterplot of PDC against Thunderstorm frequency produced.  The correlation coefficient was calculated to be 0.76.  The additional plot has been included in |
|-------------------------------------------------------------------------------------------------------------------------------------------------------------------------------------------------------------------------------------------------------------|--------------------------------------------------------------------------------------------------------------------------------------------------------------------------------------------------------------|
|                                                                                                                                                                                                                                                             | figure 2, with explanatory text added to the figure caption.                                                                                                                                                 |
| In Fig. 4 I wonder whether Rocket nose cone would be a better term than Rocket warhead.                                                                                                                                                                     | Figure 4 and its caption have been edited to change the term "warhead" to "nose cone"                                                                                                                        |
| Line 7 electric circuit, and the early evidence was found to support this model                                                                                                                                                                             | Sentence on line 7 changed to " and the early evidence that was found" to improve clarity                                                                                                                    |
| 28 its                                                                                                                                                                                                                                                      | In line 28, "it's" was corrected to "its"                                                                                                                                                                    |
| Fig. 1a. It would be better to have a photograph showing a clear gap between the metal support of the point discharge instruments and the more distant tower.                                                                                               | Figure 1a has been updated to show the mast and instruments clearly without objects in the background                                                                                                        |
| Fig. 1b. Is there a paper which describes this instrument? If so, please give a reference.                                                                                                                                                                  | A citation to Marlton et al. 2013 has been included in the caption for figure 1, as this paper describes the point discharge instrument pictured.                                                            |
|                                                                                                                                                                                                                                                             | Figure caption has been slightly altered to compensate for this.                                                                                                                                             |
| 75 Earth; however, in areas                                                                                                                                                                                                                                 | Comma replaced with semicolon in line 75                                                                                                                                                                     |
| 77. It was                                                                                                                                                                                                                                                  | Typo corrected in line 77 ("In" changed to "It")                                                                                                                                                             |
| 114 ionosphere, and restoring                                                                                                                                                                                                                               | Semicolon replaced with comma in line 114                                                                                                                                                                    |
| 282 instruments; however, they still have                                                                                                                                                                                                                   | Comma replaced with semicolon in line 282                                                                                                                                                                    |
| 285. aircraft                                                                                                                                                                                                                                               | "air" replaced with "aircraft"                                                                                                                                                                               |
| 288. It could be useful to state that this desert is in northern Mexico and southwestern USA.                                                                                                                                                               | Line 288 has been modified to clarify that
the observation site was in New Mexico,
USA                                                                                                                 |
| 308 understood, however, was                                                                                                                                                                                                                                | Additional comma added to line 308                                                                                                                                                                           |
| Acknowledgements undertaking of BM's PhD project,                                                                                                                                                                                                           | Acknowledgments modified to clarify that it was BM's PhD project.                                                                                                                                            |

**Editor Comments**

| I think the title does not reflect the historical aspect of the manuscript. Therefore the term historical may be included in the title. Something like: "The Role of Point Discharge in the historical Development"                                    | Title has been changed to "The Role of
Point Discharge in the Historical
Development of Atmospheric Electricity" |
|--------------------------------------------------------------------------------------------------------------------------------------------------------------------------------------------------------------------------------------------------------|------------------------------------------------------------------------------------------------------------------------|
| Perhaps it could be mentioned in section 2 that already in 1888 Elster and Geitel developed an instrument to measure the charge of raindrops:  Ueber eine Methode, die elektrische Natur der atmosphärischen Niederschläge zu bestimmen (Oktober 1887) | Additional sentence and citations added to line 68 to mention the contributions of Elster and Geitel                   |
| Meteorologische Zeitschrift, 5.Jg. März-
Heft, 1888, S.95-100                                                                                                                                                                                       |                                                                                                                        |
| Isn't it worth to mention Reinhold Reiter's extensive work when discussing the PG investigations? Particularly his measurements at a cable car.                                                                                                        | Additional sentence and citations added to line 55, which refers to this work.                                         |
| e.g.: Harrison and Schlegel, Hist. Geo
Space. Sci., 14, 71–75,
https://doi.org/10.5194/hgss-14-71-2023,
2023                                                                                                                                  |                                                                                                                        |
| What means the abbreviation FFAR in the caption of Fig. 4?                                                                                                                                                                                             | The caption of figure 4 has been edited to explain the meaning of FFAR (i.e Folding-Fin Aerial Rocket)                 |

**Reviewer #2 Comments**

| the end of the discussion section or beginning of the conclusions could benefit from a paragraph discussing the future potential of point discharge measurements in addition to the measurements being undertaken now. This will add to the historical context of the material already shown. | An additional paragraph has been added to the conclusions, discussing the possible direction of future point discharge investigations |
|-----------------------------------------------------------------------------------------------------------------------------------------------------------------------------------------------------------------------------------------------------------------------------------------------|---------------------------------------------------------------------------------------------------------------------------------------|
|-----------------------------------------------------------------------------------------------------------------------------------------------------------------------------------------------------------------------------------------------------------------------------------------------|---------------------------------------------------------------------------------------------------------------------------------------|

| For example, It is clear McGinnes et al 2024 is using a Point Discharge sensors so a line on their research using the sensor would be useful                                                           |                                                                                                                                    |
|--------------------------------------------------------------------------------------------------------------------------------------------------------------------------------------------------------|------------------------------------------------------------------------------------------------------------------------------------|
| L77 It instead of in                                                                                                                                                                                   | Typo corrected in line 77 ("In" changed to "It")                                                                                   |
| L127: It would be useful to define the convention of positive and negative PDC. i.e is negative PDC an outflow of                                                                                      | A convention for the polarity of PDC has been added to line 41.                                                                    |
| electrons from the point as defined in Whipple and scrace.                                                                                                                                             | Additionally, the caption for figure 4 now clarifies this meaning.                                                                 |
| Furthermore it may be prudent to add the Potential gradient convention too.                                                                                                                            | A definition of PG has been added to line 52                                                                                       |
| L189: Consider a paragraph break here. This makes it easier to see there is a shift in discussion between the positive dipole and the lower positive dipole                                            | Paragraph break inserted at line 189. Subsequent line reworked slightly to compensate.                                             |
| L266: no r in Ksanfomality                                                                                                                                                                             | Typo corrected in "Ksanformality" in line 266                                                                                      |
| L288: Import(ance)                                                                                                                                                                                     | "import" changed to "importance" in line 278                                                                                       |
| L297: Bi -polar logarithmic electrometer                                                                                                                                                               | "logarithmic electrometer" in line 297 changed to "bi-polar logarithmic electrometer"                                              |
| Equations (2, 3 and 4) Are the constants a, b and c universal values which can be used interchangeably between the equations. If not I'd suggest renaming them to different values to avoid confusion. | The constants a,b,c in equations 2,3,4 have been altered (to a,b,c,d,g,k) to make it clear that these represent distinct constants |